# Quantifying the 60-Year Contribution of Japanese Zoos and Aquariums to Peer-Reviewed Scientific Research

**DOI:** 10.3390/ani12050598

**Published:** 2022-02-27

**Authors:** Wataru Anzai, Kazuyuki Ban, Shintaro Hagiwara, Tomoya Kako, Nobuyuki Kashiwagi, Keisuke Kawase, Yumi Yamanashi, Koichi Murata

**Affiliations:** 1Research and Study Committee, Review Committee, Japanese Association of Zoos and Aquariums, 4-23-10, Taito-ku, Tokyo 110-0016, Japan; alpuruyansun@yahoo.co.jp (K.B.); hagiwara_s@animbehav-tokai.com (S.H.); t-kako@nagoyaminato.or.jp (T.K.); kashiwagi2525@gmail.com (N.K.); k_2929_k@yahoo.co.jp (K.K.); yumi.yamanashi.kycz@gmail.com (Y.Y.); murata@hama-midorinokyokai.or.jp (K.M.); 2Hiroshima City Asa Zoological Park, Dobutsuen, Asa-cho, Asakita-ku, Hiroshima-shi 731-3355, Hiroshima, Japan; 3Toyohashi Zoo and Botanical Park, 1-238 Oana, Oiwa, Toyohashi 441-3147, Aichi, Japan; 4Fukuyama Zoo, 276-1, Fukuda, Ashida-cho, Fukuyama 720-1264, Hiroshima, Japan; 5Port of Nagoya Public Aquarium, 1-3 Minato-machi, Minato-ku, Nagoya 455-0033, Aichi, Japan; 6Kagoshima City Aquarium, 3-1, Honkoushinmachi, Kagoshima-shi 892-0814, Kagoshima, Japan; 7Hitachi Kamine Zoo, 5-2-22, Miyatacho, Hitachi 317-0055, Ibaraki, Japan; 8Kyoto City Zoo, Okazaki-hosshoujicho, Sakyo-ku, Kyoto-shi 606-8333, Kyoto, Japan; 9Wildlife Research Center, Kyoto University, Tanaka Sekidencho, Sakyo-ku, Kyoto-shi 606-8203, Kyoto, Japan; 10Yokohama Zoological Gardens “ZOORASIA”, 1175-1 Kamishirane-cho, Asahi-ku, Yokohama 241-0001, Kanagawa, Japan; 11College of Bioresource Sciences, Nihon University, 1866 Kameino, Fujisawa 252-0880, Kanagawa, Japan

**Keywords:** research in zoos and aquariums, research impact, science communication, Japan

## Abstract

**Simple Summary:**

Today’s zoos and aquariums claim to be shifting from being entertainment facilities to centers for the conservation of biodiversity. To make this shift, scientific research, including understanding the biology of endangered species and improving the environment of captive animals, is essential. Several studies have examined the trends in the number of research papers published by zoos and aquariums globally and indicated that this number has increased over the past few decades. In this study, we examined the trends in the number of papers published by Japanese zoos and aquariums over the past 62 years to determine whether research activities are also developing in Japan. We found that the number of research papers has significantly increased and that the research fields have diversified since around 1990. However, we also found some problems: about a quarter of the institutions have published no papers, research targets in zoos were biased toward captive mammals and that aquariums conducted little research on animal welfare. Addressing these issues, we would argue, will help Japanese zoos and aquariums to make further progress.

**Abstract:**

With the shift in their social roles, modern zoos and aquariums are required to develop scientific research. Although zoos and aquariums worldwide have reported an increase in the number of papers they publish and the diversification of their fields in recent decades, the specific circumstances in Japan are slightly unclear. We listed peer-reviewed papers authored by Japanese zoos and aquariums using search engines and quantitatively evaluated the changes in the number of papers published over 62 years. Our results showed that papers published in Japan have increased remarkably since the 1990s, and research fields have diversified as in the rest of the world. In particular, joint research with research institutes has seen an upward trend, and the instances of English-language papers have increased. Meanwhile, the content of the research was biased. In zoos, research on animal welfare has been increasing, but the focus was heavily biased toward captive mammals. Aquariums contributed to the understanding of local ecosystems through the fundamental study of wildlife, but there were fewer papers on improving husbandry. Our results indicated that while research by Japanese zoos and aquariums is developing, research on welfare, conservation, and education regarding native endangered species must still be improved.

## 1. Introduction

In recent years, zoos have been expected to not only provide entertainment but also to play a social role in biodiversity conservation and environmental education [1,2]. To achieve these goals, it is necessary to conduct scientific research. Understanding the behavior and nutrition of each animal species should lead to improvements in animal welfare and the success of ex situ conservation, and studying the ecology of wild populations and educating citizens about them should lead to in situ conservation of endangered species. As scientific research is often presented as one of the missions of modern zoos and aquariums [3], these institutions need to take the initiative in researching a wide range of fields.

Although only a few attempts have been made to evaluate whether zoos and aquariums are contributing to conservation and education through scientific research, several studies have been reported in the past several years that have quantified the growth in the number of peer-reviewed papers [4,5,6,7,8]. According to them, the number of papers originating from zoos has increased dramatically in the past few decades, mainly from institutions based in North America, Europe, China, and South Africa [8]. The main focus of the papers used to be applied research that would improve basic husbandry, such as veterinary science and breeding reports, but nowadays, the research fields have diversified and studies in fundamental biology are also increasing [4,5,6,7,8]. Especially in recent years, the improvement of animal welfare and the conservation of endangered species in captivity have emerged as major keywords [8]. It is considered that shifts in public and industry perceptions of animal welfare and the biodiversity crisis have been reflected in the scope of published papers from zoos and aquariums.

However, it is difficult to determine whether zoos and aquariums are contributing to conservation, as those studies lack some perspectives. First, it is unclear whether the research is led by the zoos. Sometimes zoos become one of the authors simply by providing samples to outside biologists, but if this is the case for most of the published papers from zoos, it cannot be regarded as a substantial contribution by them. This is one way to evaluate whether the first author belongs to zoos, but none of these studies mention it. Second, it is not clearly stated whether the research targets are captive or in the wild. Although important findings can be obtained from research on captive animals, the extent to which research is conducted on wild populations is considered to be an important indicator to evaluate whether the goal of biodiversity conservation is being achieved. Finally, these studies did not count papers in local languages. They used journal databases [4,5,6,7,8], most of the papers covered were in English. In non-English-speaking countries, many papers are published in native languages, and in some cases, the local language is given priority, especially for studies on the conservation of the local ecosystem [9]. This is the case in Japan; several conservation biology studies by zoos or aquariums have been published in Japanese journals, including one published by the Japanese Association of Zoos and Aquariums (JAZA). One such example is the conservation of the Japanese giant salamander (*Andrias japonicus*), a species endemic to Japan, by Asa Zoological Park. Based on the breeding ecology they discovered in their field research [10], they provided advice to the local government on river improvement projects in the habitat of the rare species and installed an artificial den that allows the giant salamander to breed naturally even on paved river banks [11]. This zoo-led in situ conservation project was reported in the *Journal of Japanese Association of Zoos and Aquariums* (JJAZA), an original peer-reviewed journal published by the JAZA, which meant other local governments or conservation organizations in Japan have been able to utilize it, but it has been missed by the international research database. In quantifying the expansion of research activities in zoos and aquariums, it is important to include these articles rendered in the local language.

Based on these points, we clarify the shift in the number of peer-reviewed papers by Japanese zoos and aquariums. Although a declining trend was reported in the number of papers in the JJAZA [12], the overall trend in Japan remains unclear. To find out whether Japanese zoos and aquariums are contributing to conservation and education through their research efforts, it is important to compare the local status quo with the worldwide trend and to identify problems and gaps. Our goal in this paper is to understand the history and current status of research activities in Japanese zoos and aquariums. Therefore, we compiled 60 years of peer-reviewed papers authored by JAZA members to explore how the number and subjects of publications have shifted over the years.

## 2. Materials and Methods

The research contribution of each of the 140 active JAZA member institutions, 90 zoos and 50 aquariums, extant on 1 April 2021, was quantified. We listed peer-reviewed papers in which one or more authors belonged to those JAZA institutions. Specifically, we intended to include original papers, review articles, case reports, short communications, and technical materials published in peer-reviewed scientific journals, including the JJAZA. Non-peer-reviewed papers such as bulletin papers published by universities or museums, dissertations, conference proceedings, Special Issues, book reviews, and book chapters were not counted in the total. Papers published between 1959, when the JJAZA was founded, and 2020, were listed.

Previous studies have used databases such as the Thomson Reuters Web of Science or Elsevier Scopus [4,5,6,7,8], but they do not cover many Japanese academic journals, therefore we used three journal search engines: PubMed, Google Scholar, and J-STAGE (an electronic journal platform in Japan, managed by the Japan Science and Technology Agency). In PubMed and J-STAGE, we used the advanced search function and searched by the name of each institution as an author affiliation. In Google Scholar, we checked all search results with the institution name in double quotation marks and extracted the target literature. In all cases, the search was conducted using the Japanese name of the institution, the old Japanese name, and the English name. In the JJAZA, we selected all the target articles from all the past publications because the journal was available only among JAZA members, and it cannot be found by these search engines, with only a few exceptions. All the articles selected were carefully reviewed and those that were duplicates or did not fulfill the criteria were excluded. Data were collected between May and September 2021.

For all articles, the following elements were recorded: year of publication, institution, first author’s affiliation, language, research target, and research field. The affiliation of the first author was recorded as being a JAZA member or not; this indicates whether the paper is authored mainly by zoo and aquarium staff or co-authored with another institution. The language of the article was identified as Japanese or English. Research targets were classified as captive animals, wild animals, both, or others (Table 1). “Wild animals” refers to field surveys or studies using collected wild individuals or bycatch. “Others” refers to studies on plants, visitors, facilities, etc. In addition, studies targeting animals were recorded separately by their taxonomic groups, mammals, birds, reptiles, amphibians, fish, invertebrates, or several of these. For the research field, we listed 14 field keywords (Table 2; Animal care, Animal welfare, Behavior, Conservation, Ecology, Education, Exhibition, Genetics, Morphology, Nutrition, Physiology, Reproduction, Taxonomy, and Veterinary science) related to zoo and aquarium research and selected all the relevant keywords or none for each article. For instance, in a study that estimates the breeding cycle by behavioral observation and hormone measurement, Behavior, Breeding, Physiology, and Reproduction were chosen as field keywords. The annual variation was illustrated for each of the above factors.

## 3. Results

### 3.1. Overall Trends and Affiliation of the Author

We compiled 2789 papers from 103 institutions. The mean per institution was 21.23, and the median was 11. There was a wide variation among institutions, with the top four accounting for 33% of the total, while 37 (27 zoos and 10 aquariums) had no publications at all. The overall number of papers from the JAZA showed an increasing trend (Figure 1). The growth was especially remarkable after 2000 and did not match the growth curve of JAZA member institutions. In terms of the affiliation of the first author, the number of co-authored papers increased significantly from around 1990, from fewer than 10 papers in 1990 to nearly 100 papers in 2020, more than double the number of papers first-authored by JAZA staff. The number of first-authored papers by JAZA staff increased slightly in recent years, though this has fluctuated over 60 years.

The trends of results differed between 63 zoos and 40 aquariums. On the one hand, in zoos, the number of first-authored papers had decreased compared to the peak year of 1967. However, the number of co-authored papers had increased, and the recent overall trend was upward. On the other hand, in aquariums, the number of papers had increased rapidly from the 1990s, although it had been low until then. Both first- and co-authored papers continued to increase until 2020, despite the decrease in member institutions since 2005.

### 3.2. Language

The transition of the number of papers in English and Japanese is illustrated in Figure 2. It shows that almost all papers were written in Japanese until 1988, but papers in English increased in number after that. Since around 2005, more literature has been published in English than in Japanese.

In the results for zoos only, the ratio of papers written in English began to increase around 1995. For the past 10 years, the number of studies in English and Japanese has been almost the same. Aquariums published a more notable number of papers in English than zoos in recent years, nearly double the number in Japanese.

Additionally, the number of first-authored papers in English by JAZA staff is shown in Figure 3. The number of papers in English by aquarium staff has been increasing gradually since around 1990 and has remained stable thereafter, whereas papers in English written by zoo staff were fewer than five per year for a long time but began to increase in the 2010s.

### 3.3. Research Targets

Figure 4 shows the annual changes in the number of papers sorted by research target. Although the total number of papers from zoos had been increasing, most of them have focused on captive animals throughout the survey period. Furthermore, more than 60% of them dealt with mammals. About 23% of papers were on birds, but there were very few on reptiles, amphibians, and invertebrates.

In aquariums, research on captive fish had been the main focus, but the target has diversified since 2000. While studies targeting both captive animals and “others” are increasing, the growth in the number of papers on wild animals was more pronounced, exceeding the total number of studies on captive animals. Table 3 lists 771 papers from both zoos and aquariums targeting wild animals and indicates that 73%, or 578 papers, focused on native Japanese species. Regarding taxonomic groups, mammals, fish, and invertebrates had all been targeted well. The studies focusing on reptiles and amphibians were as scarce as in zoos.

### 3.4. Research Fields

Initially, most research highlighted issues that had some association with the basic husbandry of zoo and aquarium animals, such as Animal care, Reproduction, and Veterinary science (Figure 5). However, since around 2000, the study fields have diversified gradually. In zoos, although “zoo-related” research has dominated for all the 60 years, genetics and physiology have increased as study topics since around 2000. In the 2010s, papers about welfare began to appear. Education, Exhibition, and Nutrition were still seldom studied; for each of these fields, the number of papers had been fewer than 50 in whole 62 years.

A limited amount of the published literature about Animal care and Ecology came from aquariums in the 1900s. However, since 2000, the number of studies in Ecology, Morphology, and Taxonomy has increased greatly compared to other fields. As in zoos, studies in Genetics, Physiology, and Reproduction were also on the increase, but there was almost no research on Animal welfare. Further, similar to zoos, there were few papers on Education, Exhibition, and Nutrition.

## 4. Discussion

Our analysis over 60 years showed that JAZA zoos and aquariums have increasingly contributed to scientific research through the publishing of peer-reviewed papers. According to Escribano et al. [8], over the past 16 years, the production of scientific papers by zoos has increased much faster than the average rate of all scientific papers. Statistics from Japanese government agencies show that the total number of all research papers in Japan in 2019 was 3.3 times higher than that in 1981 [13], whereas the number of JAZA papers has increased nearly sevenfold (Figure 1). One of the many reasons for this could be the increase in the number of member institutions in the past 60 years. However, since the number of papers increased faster than the number of member institutions (Figure 1), the research activities of the JAZA seem to be progressing as with zoos around the world. Nevertheless, since the top four institutions account for 33% of the total number of papers, there appears a large bias in the number of publications among the institutions. With 37 institutions publishing no papers, the simple average number of publications per year per institution is less than one. The number of publications was also unbalanced in other regions. In the European Association of Zoos and Aquaria (EAZA), only seven institutions accounted for 37% of the total peer-reviewed papers published by 291 zoos over 21 years [7]. Although this number is gradually increasing in the JAZA, the overall level must grow.

This is similar to the global trend in that physiological and genetic methods are becoming more common as biology develops [8], and the number of papers on these methods is increasing and diversifying JAZA’s research (Figure 5). However, papers by the JAZA were still biased toward “classical zoo and aquarium-related” research, such as animal care, reproduction, and veterinary science. As the main goals of the World Association of Zoos and Aquariums are to improve animal welfare, promote environmental education, and encourage global conservation [14], the fact that the papers published in recent years from zoos and aquariums worldwide have focused on animal welfare science or conservation biology is a change in line with these goals [8]. In these fields, which are important for zoos today, the JAZA seems to be far behind the rest of the world. In 63 zoos of the JAZA, “classical research” has dominated published papers, past and present. Nevertheless, research on welfare and behavior has seen an increasing trend in recent years, so it seems that attempts to improve captive environments based on scientific assessments are progressing gradually. A further task is to resolve the fact that the research target is heavily biased toward captive mammals (Figure 4). Studies on reptiles and amphibians were fewer than those on mammals and birds, although this seems to be the common case worldwide [5,6,8]. Given the goal of modern zoos to contribute to wildlife conservation, more studies should focus on endangered species [15]. However, it has been pointed out that more papers on mammals, especially large charismatic species that are popular with the public, are influenced by public and media interest trends [16]. Improving the welfare of captive, non-mammal/avian species, conservation research on wildlife, and the evaluation of education or exhibition for visitors are gaps that must be filled as well as research topics that JAZA zoos should focus on in the future.

In contrast, most papers published by JAZA aquariums were descriptive studies of taxonomy, ecology, and morphology (Figure 5), many of which focus on wild fish or invertebrates and most of which examine native Japanese species (Table 3). These studies contribute to the understanding of the ecosystem of wildlife, particularly the populations in Japan. Japanese aquariums are considered to be making significant contributions to basic biology and understanding of biodiversity in Japan. As in the case in zoos, there were only a few papers on reptiles and amphibians, although a large number of descriptive studies of fish have been published (Figure 4). This seems to be a unique aspect compared to the results of the small number of papers dealing with fish in zoos and aquariums around the world [5,6,8]. One of the reasons seems to be that ichthyology and fisheries are flourishing in Japan, a country surrounded by the sea, and many college graduates who major in ichthyology or fishery science are employed by aquariums [17]. In contrast, there was even less research in fields such as animal welfare and nutrition than in JAZA-affiliated zoos. Improving the care of captive animals is a critical problem that Japanese aquariums need to face.

Most papers published until around 1990 were written in Japanese, but the number of English-language papers increased gradually, and in recent years, the number of papers in English has exceeded those in Japanese (Figure 1). To disseminate the outcomes to researchers around the world, research results should be published in international journals in English [18]. The fact that the number of papers from the JAZA that were published in English is increasing is a positive change, and there are three main contributory factors to consider. First, there is an increase in collaborative research with universities and other research institutions. Zoologists have perceived Japanese zoos and aquariums as entertainment facilities and kept them at arm’s length [19], yet the relationship between the two has been improving, reflected in the remarkable increase in the number of co-authored papers since around 1990 (Figure 2). English papers and co-authored papers show a similar increasing trend, indicating that papers by research institutions were often submitted to international journals. It should be noted that these are not zoo- or aquarium-led projects; nevertheless, it is an important progression that even institutions with limited ability to disseminate research are beginning to contribute to the publication. The second factor is the opening of two aquariums with research departments. Traditionally, many of JAZA’s member facilities neither possess a research department nor do they employ curators or researchers, hence the fewer first-authored papers. However, Lake Biwa Museum, which opened in 1996, and the Okinawa Churaumi Aquarium, which opened in 2002, have research departments and have published many papers. Lake Biwa Museum has published 420 papers (14% of all papers in this study) and the Okinawa Churaumi Aquarium has published 232 (8%). The rapid increase in publishing by JAZA aquariums since the 2000s is considered to be largely due to the contribution of these two facilities. Nevertheless, focusing on these institutions with the largest number of publications, the Lake Biwa Museum peaked in 2007 and has been slowing down (Appendix A). The Ueno Zoological Gardens and Tama Zoological Park, which have the third- and fourth-highest numbers of papers overall, respectively, did not show an increasing trend in recent years, although they have achieved a great deal by publishing papers for 60 years (Appendix A). Moreover, several other zoos have created new, full-time research positions in the past few years, and publications from them may be expected to increase in the future. In short, the two aquariums have contributed largely to the growth since 1990, but one can seem to assess that more institutions have become committed to research in the last decade. The last factor is a change in the organization and the mindset of the staff. Even in facilities that have no full-time researchers, other staff (keepers, veterinarians, or educational staff) has increasingly been writing research articles. At their inception, Japanese zoos did not have many employees with degrees other than veterinarians, and many directors of zoos were also veterinarians, but in recent years, the number of staffs with university or higher degrees and with curatorial qualifications has been increasing [17,19]. As the number of papers written in English by the staff of zoos or aquariums is also increasing (Figure 3), even at institutions that do not employ full-time researchers, e.g., [20,21], it can be concluded that the efforts of many staff are supporting JAZA’s international research challenge.

More than half the total papers were in Japanese. Most organizations involved in local ecosystem conservation and the citizens who are our targets of environmental education are native language speakers. Scientific papers written in English may limit the ability to communicate important results to local practitioners and decision-makers, such as environmental managers [18]. Indeed, 35% of the scientific papers on biodiversity conservation published worldwide in 2014 were in non-English languages [9]. More than 70% of JAZA’s studies on wild animals targeted native species, and there were more papers in Japanese than in English especially in the fields of ecology and conservation (Table 3). Writing in home languages is also a valid option for results that should be applied to local conservation activities. One of the roles of zoos and aquariums should be to continue to publish their research about native species in their native language, which can be expected to have an environmental–educational benefit to the local community. However, many Japanese-language journals published by academic societies in Japan are not registered in international databases, such as Scopus or Web of Science. Some previous studies have used these superior databases to evaluate the number of articles by zoos [4,5,6,7,8], but this method may have missed some papers in local languages in each country. Although those papers may not be regarded as outstanding in an international context, e.g., [10,11], it is important to reflect on the number of such local papers when quantitatively evaluating the contribution of zoos and aquariums to society.

## 5. Conclusions

The social role of zoos and aquariums around the world is changing to that of conservation and environmental education, and the increasing number of peer-reviewed papers may be evidence of their contributions through scientific research. As we have shown, in Japan, JAZA’s scientific research has developed positively. However, this has also clarified that the JAZA is lagging behind the rest of the world in research fields that could contribute to modern goals for the zoos. Most studies by JAZA zoos focused on captive mammals, and it seems that more work needs to be conducted on conservation and improving the welfare of endangered species in other taxonomic groups, though it is also a problem worldwide. Promoting research on animal welfare and behavior should provide a useful scientific background for the formulation of the husbandry guidelines that the JAZA is currently working on. In contrast, the negligible number of welfare papers published is a serious problem for JAZA aquariums. The relative ease of collecting individuals in fishery-rich Japan may have helped promote aquarium research on wildlife while neglecting efforts to care for captive individuals. Discussions on the welfare of domestic aquatic species in the JAZA are still in their infancy; for instance, while a large-scale study is developing worldwide to improve the welfare of cetaceans [22], breeding efforts in Japan do not seem to reach the scientific research level. Therefore, several Japanese institutions need to conduct cooperative studies both in the country and overseas, and should publish them globally to deepen those discussions rapidly. Although it remains in doubt that the JAZA can be labeled as an institution for conservation research, as it has many members with no publications in the past 62 years, filling the gaps identified in this study will lead to a shift in roles. Shortly, it will be necessary to establish a framework to encourage scientific research at each member facility, such as forming a research committee within the JAZA organization and evolving and contributing to the promotion and protection of biodiversity and social education by developing more scientific activities.

## Figures and Tables

**Figure 1 animals-12-00598-f001:**
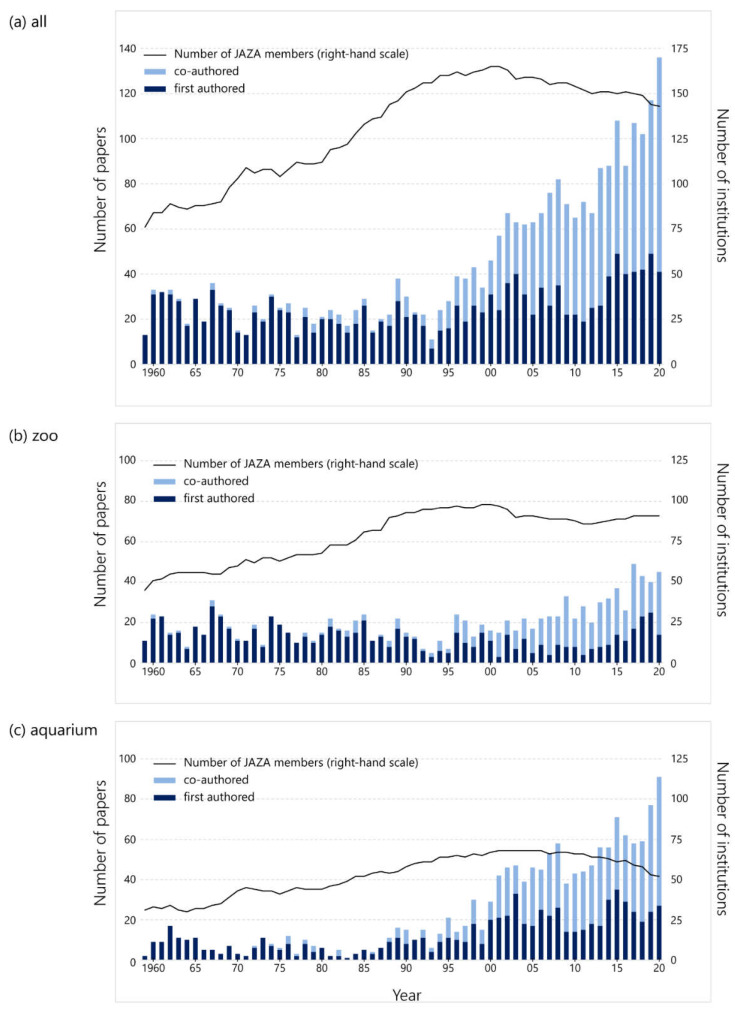
The annual trend in the number of papers colored by the affiliation of the first author. Dark blue bars indicate JAZA staff as the first author, and light blue indicates outside researcher as one. The line represents the number of JAZA member institutions. (**a**) All JAZA members; (**b**) zoos; (**c**) aquariums.

**Figure 2 animals-12-00598-f002:**
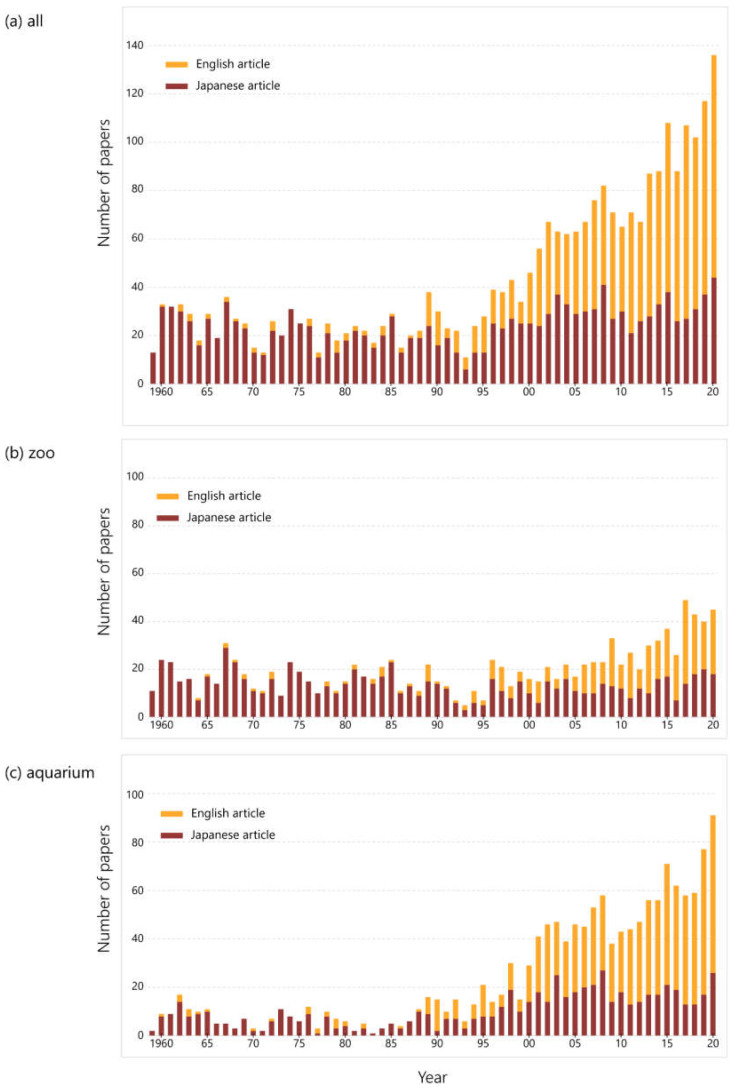
The annual trend in the number of papers colored by language. Dark red bars indicate articles written in Japanese, and orange are in English. (**a**) All JAZA members; (**b**) zoos; (**c**) aquariums.

**Figure 3 animals-12-00598-f003:**
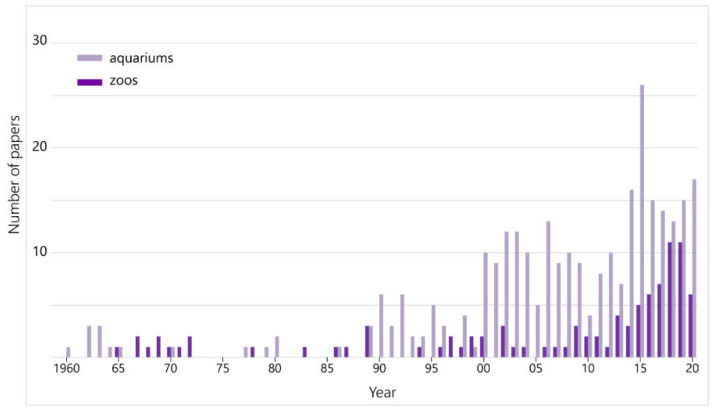
The annual trend in the number of English papers first-authored by JAZA staff. Dark purple bar indicates papers by zoos, and light indicates by aquarium staff.

**Figure 4 animals-12-00598-f004:**
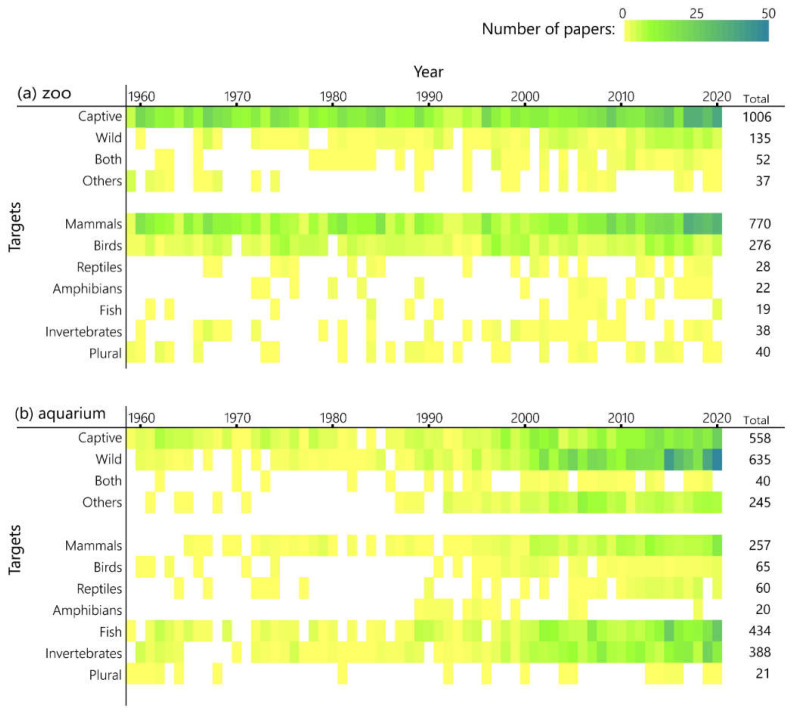
Research fields by year. Color scale indicates the number of papers selected as being appropriate for keywords in the year. (**a**) zoos; (**b**) aquariums.

**Figure 5 animals-12-00598-f005:**
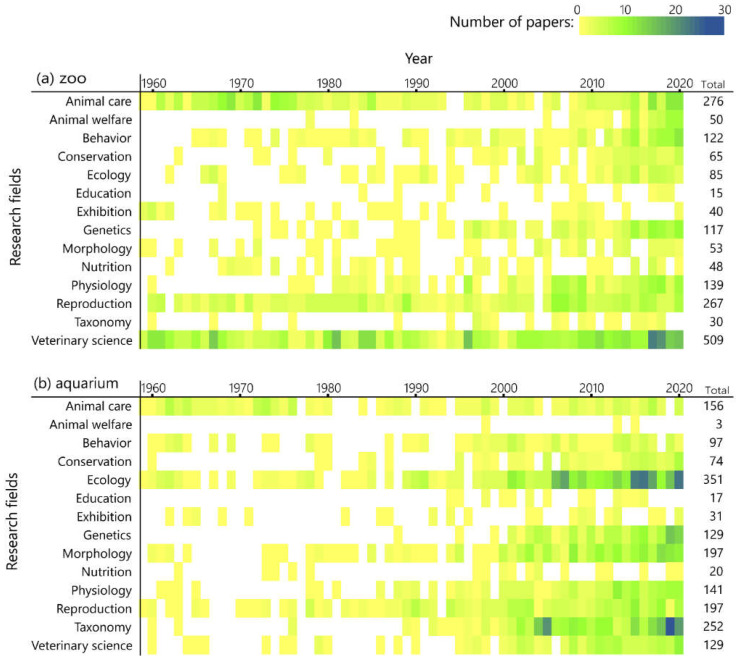
Research targets by year. Color scale indicates the number of papers targeting each category in a given year. (**a**) zoos; (**b**) aquariums.

**Table 1 animals-12-00598-t001:** List of classifying research targets.

Research Target	Research Target—Taxonomic Groups
Captive animals	Mammals
Wild animals	Birds
Both; captive and wild	Reptiles
Others; e.g., plants, visitors, facilities	Amphibians
	Fish
	Invertebrates
	Plural; several of above

**Table 2 animals-12-00598-t002:** Field keywords and examples used in our analysis.

Field Keywords	Examples
Animal care	rearing technique, artificial rearing, basic husbandry
Animal welfare	enrichment, quality of life assessment, stereotypies
Behavior	behavioral observation, ethograms
Conservation	conservation biology, prevention of alien species
Ecology	description of biological traits, biogeography
Education	educational effects for the visitor, teaching materials
Exhibition	displaying method, specimen preparation
Genetics	studies dealing with DNA, karyotype analysis
Morphology	morphometrics, comparative anatomy
Nutrition	diet survey, nutritional analysis
Physiology	endocrinology, enzymology
Reproduction	report of breeding, reproductive physiology
Taxonomy	new species description, systematics
Veterinary science	case report, parasitology

**Table 3 animals-12-00598-t003:** The detailed number of papers targeting wild animals. The number of papers targeting wild animals is illustrated separately by research field (show only the four most frequent, and total), language, and whether the target is a Japanese native species or not. In the case of native species, the taxonomic groups are also shown.

Fields	Conservation		Ecology		Morphology		Taxonomy		All Papers	
Language	Jap	Eng	Jap	Eng	Jap	Eng	Jap	Eng	Jap	Eng
Non-native species	8	14	11	43	2	37	7	98	31	190
Native species	38	33	144	103	34	67	34	108	260	318
Mammals	4	8	16	18	3	3	0	1	32	37
Birds	11	7	33	8	3	1	0	0	52	35
Reptiles	2	4	7	14	1	2	2	5	8	25
Amphibians	4	0	7	0	2	0	0	1	18	2
Fish	6	10	47	38	9	21	11	26	75	102
Invertebrates	8	2	33	23	16	40	21	74	71	114

Jap: written in Japanese, Eng: in English.

## Data Availability

The data presented in this study are available on request from the corresponding author.

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
