# Peer review of "Quantifying the 60-Year Contribution of Japanese Zoos and Aquariums to Peer-Reviewed Scientific Research"

_animals, 2022, doi:10.3390/ani12050598_

Round 1

Reviewer 1 Report

This study presents an in-depth assessment of peer-reviewed scientific publications by Japanese zoos and aquaria. The article falls in line with other recent publications which aim to assess the contribution by ex-situ conservation facilities worldwide.

The authors have developed a solid approach and present the data in clear graphics, drawing sound conclusions.

The only downside is that the authors fail to address the massive inequality in contributions across the JAZA institutions (22% of the papers appear to be from only 2 institutions, and close to one third of the institutions registered no publications). The potential bias induced by these high-publishing institutions needs to be addressed; do they skew the trends in type of publications, language, or other aspects?

The authors need to acknowledge this telling inequality (which is not exclusive to Japan), and if possible provide whatever data might help explain the failure by many zoos and aquaria to publish.

Author Response

Thank you for your review of our manuscript. We carefully examined your comments and revised the manuscript. Our responses to your points are as follows:

The only downside is that the authors fail to address the massive inequality in contributions across the JAZA institutions (22% of the papers appear to be from only 2 institutions, and close to one third of the institutions registered no publications). The potential bias induced by these high-publishing institutions needs to be addressed; do they skew the trends in type of publications, language, or other aspects?

The authors need to acknowledge this telling inequality (which is not exclusive to Japan), and if possible provide whatever data might help explain the failure by many zoos and aquaria to publish.

[Response] Following your suggestion, we have added the mention of the inequality between institutions. We have specified in the results that the mean and median for the total number of publications, that the top publishing institutions account for a large proportion of overall papers, and that 37 institutions have no publications (lines 162-165). We have also made an additional discussion based on these results (lines 254-260).

 Then shift in the number of papers focusing on the top two aquariums and two zoos is shown in new figures (S1, S2), and their influence on the overall trend is also discussed (lines 321-330).

Reviewer 2 Report

Quantifying the 60 years contribution of Japanese zoos and aquariums to peer-reviewed scientific research

In the above-named review, the authors describe and quantify the development of research publications from Japanese zoos and aquariums.

In short, I find this manuscript very well written and informative. I have only one comment, which is a question really.

GENERAL COMMENTS

In my work, I do a lot of hypothesis-driven research. Testing for statistical significance in my target variable between groups is a big part of it. Coming from this background I find it very unusual to see such descriptive work where data is handled without any statistical significance testing. And I am wondering if it could make sense here, to include statistical back up showing, e.g., that there is a significant publication bias towards specific taxonomic groups or research areas. On the other hand, it seems unnecessary to statistically show that the number of publications and the proportion of English publications significantly increased over the previous years.

Once again, this more like a question or side note and I am curious to hear the journal’s and authors’ opinion on this comment rather than thinking that this needs to be done.

SPECIFIC COMMENTS

Line 51: What issues would that be?

Line 173: “colored by the affiliation of the first author” it is, I guess?

Author Response

Thank you for your review of our manuscript. We carefully examined your comments and revised the manuscript. Our responses to your points are as follows:

In my work, I do a lot of hypothesis-driven research. Testing for statistical significance in my target variable between groups is a big part of it. Coming from this background I find it very unusual to see such descriptive work where data is handled without any statistical significance testing. And I am wondering if it could make sense here, to include statistical back up showing, e.g., that there is a significant publication bias towards specific taxonomic groups or research areas. On the other hand, it seems unnecessary to statistically show that the number of publications and the proportion of English publications significantly increased over the previous years.

Once again, this more like a question or side note and I am curious to hear the journal’s and authors’ opinion on this comment rather than thinking that this needs to be done.

[Response] We don’t ignore the importance of statistical analysis, but we believe it is not necessary for our results.

The first reason is that it is not in line with our argument. Since our goal is to qualitatively understand the current stats and gaps, it is not important whether the numerical differences between the groups are statistically significant or not. While we could test for bias in the total number of papers between fields, for instance, we don’t consider that to be critical to our argument.

Secondly, we don’t have the appropriate statistical methods to track the time series shifts. Probably for the same reason, previous studies that have explored the shift in the number of papers in the world have not conducted a statistical analysis [4-8, references], and this is also the reason why we do not conduct statistical tests. A further amount of data is needed for a more concrete and statistical discussion.

Line 51: What issues would that be?

[Response] We have rewritten this section to be more specific about what the “issues” is (lines 50-52).

Line 173: “colored by the affiliation of the first author” it is, I guess?

[Response] Corrected as pointed out (line 180).

Reviewer 3 Report

The authors want to elucidate whether research activities are (as in other parts of the world) also developing in Japanese zoos and aquaria. For that they evaluate the increase in peer-reviewed publications authored or co-authored by these institutions, deliberately counting in publications in Japanese language. The aim itself is interesting, as there isn't any data on this available, yet. However, the kind of analysis presented is very limited. Most importantly, the authors tackle a question that is not purely quantitative in scope (the contribution of Japanese Zoos and Aquaria to scientific research and conservation/education) by purely quantitative methods (counting the number of publications). Their conclusions are not in range of their data and they are not normatively neutral: They clearly interpret their finding of an increase in papers in a qualitative way in the sense that it constitutes a "progress" (to scientific knowledge and to education and conservation) and that Japanese institutions are developing in the right direction. By that they adopt very uncritically the position of zoo umbrella organizations like JAZA and WAZA who support the ethical justification of their institutions by claiming to move towards conservation and education. As there is a complex scholarly debate about such strategies, the findings of the authors would ask for discussion and for more careful interpretation. Instead the authors arrive at conclusions that are not argued for and in fact not supported by their findings (they say, for example, that especially the publications in Japanese language contribute to the conservation of local ecosystems. This specific contribution, however, is not properly addressed in their analysis. The criteria that ensure that the papers make a valuable contribution are not given, nor have they been tested for in their analysis).  The gaps and biases that become visible by help of their evaluation are named but not discussed (e.g., a bias in research towards mammals, the fact that only two instutitions out of 103 make up for almost a quarter of all publications, the fact that it is mainly research cooperations that increase etc.). Instead, the authors stick to their rather positive conclusions for Japanese zoos and quaria in general.

I have left many comments in the text where these main points, as well as many other related points, become visible. I strongly recommend to the editor(s) to have a look at those comments to understand the scope of the problems.

The fact that the authors write as "We, the JAZA"(see conclusions) supports my strong feeling that this paper rather shows the quality of a JAZA report, following the political positions of JAZA, and not the quality of a critical, in-depth analysis starting from a neutral vantage point, something one would expect for a scientific paper. This is unfortunate because the authors could have used the limits of their data to actually open up or continue a very insightful debate.

Author Response

Thank you for your review of our manuscript. We carefully examined your comments and revised the manuscript. Following the Editor’s suggestion, we didn’t deal with some of the comments, but we tried to make a critical and neutral argument as you pointed out. In particular, we have explained previous examples of zoo and aquarium research targeting wildlife conservation, and have also detailed the results of these studies (Table 3). Please see the comments in the attached PDF.

Round 2

Reviewer 3 Report

I appreciate the authors trying to include my comments. The many points that remain unaddressed are in the responsibility of the editor.

I recommend to rephrase two sentences from the conclusions:

In lines 370-375 (conclusions) the authors now write "Discussions on the welfare of domestic aquatic species are still in their infancy; for instance, while views on cetacean husbandry are changing, they have not yet reached the scientific research level. Therefore, it is necessary to deepen these discussions based on a more biological background."

It is simply not true that discussions on the welfare of domestic aquatic species (and in specifically on the welfare of cetaceans in captivity) haven't reached scientific research level. Accredited zoos around the world are  increasing research there precisely BECAUSE they are aware of the challenges of keeping those mammals in zoos and aquaria. See e.g. (and this is just the research output in PLoS ONE): https://collections.plos.org/collection/cetacean-welfare/

Author Response

Thank you for your quick response. Following your insightful suggestions, we have rephrased our conclusion about the contrast between the problems in Japan and the outcomes in the world.